# Direct Derivatization in Dried Blood Spots for Oxidized and Reduced Glutathione Quantification in Newborns

**DOI:** 10.3390/antiox11061165

**Published:** 2022-06-14

**Authors:** Isabel Ten-Doménech, Álvaro Solaz-García, Inmaculada Lara-Cantón, Alejandro Pinilla-Gonzalez, Anna Parra-Llorca, Máximo Vento, Guillermo Quintás, Julia Kuligowski

**Affiliations:** 1Neonatal Research Group, Health Research Institute La Fe, 46026 Valencia, Spain; isabel_ten@iislafe.es (I.T.-D.); alvarosogar@gmail.com (Á.S.-G.); inmaculada_lara@iislafe.es (I.L.-C.); alejandro_pinilla@iislafe.es (A.P.-G.); parra.ann@gva.es (A.P.-L.); maximo.vento@uv.es (M.V.); 2Division of Neonatology, University and Polytechnic Hospital La Fe, 46026 Valencia, Spain; 3Health and Biomedicine, LEITAT Technological Center, 08225 Terrassa, Spain; 4Unidad Analítica, Health Research Institute Hospital La Fe, 46026 Valencia, Spain

**Keywords:** redox status, oxidative stress, dried blood spot, liquid chromatography–mass spectrometry (LC-MS)

## Abstract

The glutathione (GSH)-to-glutathione disulfide (GSSG) ratio is an essential node contributing to intracellular redox status. GSH/GSSG determination in whole blood can be accomplished by liquid chromatography–mass spectrometry (LC-MS) after the derivatization of GSH with *N*-ethylmaleimide (NEM). While this is feasible in a laboratory environment, its application in the clinical scenario is cumbersome and therefore ranges reported in similar populations differ noticeably. In this work, an LC-MS procedure for the determination of GSH and GSSG in dried blood spot (DBS) samples based on direct in situ GSH derivatization with NEM of only 10 µL of blood was developed. This novel method was applied to 73 cord blood samples and 88 residual blood volumes from routine newborn screening performed at discharge from healthy term infants. Two clinical scenarios simulating conditions of sampling and storage relevant for routine clinical analysis and clinical trials were assessed. Levels of GSH-NEM and GSSG measured in DBS samples were comparable to those obtained by liquid blood samples. GSH-NEM and GSSG median values for cord blood samples were significantly lower than those for samples at discharge. However, the GSH-NEM-to-GSSG ratios were not statistically different between both groups. With DBS testing, the immediate manipulation of samples by clinical staff is reduced. We therefore expect that this method will pave the way in providing an accurate and more robust determination of the GSH/GSSG values and trends reported in clinical trials.

## 1. Introduction

Oxidative stress is of essential importance in the physiology of the fetal-to-neonatal transition and plays a key role in the pathophysiology of a range of conditions observed in term and preterm infants coined as the “free radical disease of the newborn” [1]. Glutathione (GSH) is a ubiquitous tripeptide that comprises a wide range of functions in metabolism through its reducing properties, detoxification action, and its role in the synthesis of bioactive lipids and iron–sulfur clusters as well as oxidative protein folding and redox signaling [2]. Together with oxidized glutathione (GSSG), the redox couple (GSH/GSSG) is a major determinant node contributing to the cytoplasmic redox potential [3] and its equilibrium is indispensable for adequate cell reproduction, growth, and differentiation [4]. In addition to its antioxidant properties, GSH also functions as a cysteine reservoir and is broken down in response to the shortage of cysteine, thus compromising protein synthesis [5]. Under oxidative stress conditions, e.g., during the fetal-to-neonatal transition, the consumption of GSH is increased and GSSG is eventually secreted into circulation. However, in the context of prematurity, despite the suboptimal performance of some antioxidant defense systems [6], the reported GSH concentration in the cord blood of preterm infants at birth exceeds that of term infants [7]. The GSH/GSSG redox couple has been profoundly studied in the context of newborn resuscitation. While the use of oxygen has contributed dramatic reductions in mortality, oxidative stress-related diseases such as the retinopathy of prematurity, bronchopulmonary dysplasia or intracranial hemorrhage have become a major concern [8,9]. In this context, significantly lower levels of GSH/GSSG at 28 days of life were linked to protracted oxidative stress in infants resuscitated with an inspired fraction of oxygen (FiO_2_) of 1 vs. 0.21 [10]. This culminated in today’s recommendation to initiate ventilation with an FiO_2_ of 0.21 in term infants and between 0.21 and 0.30 in preterm infants [11].

GSH quantification in whole blood is challenged by the complexity of the blood matrix as well as the rapid deterioration of the sample, including enzymatic- and non-enzymatic GSH oxidation [12,13,14,15,16]. Furthermore, GSH levels were proposed as a surrogate for assessing the time-dependent degradation of blood samples prior to plasma extraction, as even minor leakages of GSH from erythrocytes lead to an increase in plasmatic GSH levels [17]. Whilst GSH concentrations in blood plasma ranging between 2 and 20 μM have been reported in the literature, in whole blood, they are notably higher (mM range), as over 95% of GSH is located inside the erythrocytes [18]. Current state-of-the art involves the simultaneous determination of GSH and GSSG by stable isotope dilution liquid chromatography–tandem mass spectrometry (LC-MS/MS) [19,20,21,22,23]. Alternative approaches rely on electrophoretic separation, the use of hyphenated systems with different detectors, enzymatic assays, or electrochemical determinations [1]. A rapid method for the direct determination of GSH in whole blood samples by surface-enhanced Raman spectroscopy (SERS) aiming at the implementation of a bed-side approach to avoid sample storage and hence, overcome issues related to samples stability, was developed [24]. This method was rendered suitable for accurate GSH quantification in only 2 µL of blood from term newborns, but it did not enable the simultaneous determination of GSSG. On the other hand, the most commonly used approach to prevent the formation of oxidation artifacts makes use of the Michael addition reaction for the alkylation of the thiol group of GSH with *N*-ethylmaleimide (NEM) [25]. However, evidence supports that due to the lack of stability of GSH, the accurate analysis of GSH/GSSG in blood is only possible if the derivatization step is carried out immediately after sample collection [19]. While this is feasible in a laboratory environment, its implementation in the clinical scenario is cumbersome. Thus, differences in the sample collection and derivatization steps across clinical studies lead to ranges that differ substantially, even in similar populations. Although a range of different methods for the quantification of GSH and GSSG were reported, relevant limitations regarding sample collection and storage suitable for their use in a clinical environment, and especially the neonatal intensive care unit and the delivery room can be identified, including (i) the use of relatively large sample volumes (typically tens to hundreds of microliters of whole blood); (ii) time and instrumentation (e.g., centrifuges) required for sample pretreatment; (iii) access to an ultra-freezer for cold sample storage; and (iv) the need for experienced personnel.

Small blood volumes can be collected and analyzed employing dried blood spots (DBS). A drop of whole blood obtained from a heel, toe, or finger prick with a single-use safety lancet is deposited onto a filter card and left to dry before being sent to the laboratory for analysis [26]. Depending on the application, DBS cards enable long-term storage without the need of ultra-low temperatures as typically required for the storage of blood samples [27]. For unstable metabolites, DBS on-card derivatization was proposed as a reliable alternative for quantitation purposes [28,29]. On the other hand, variation due to changes in the blood hematocrit can be remediated by the collection of accurate volumes of blood and commercial devices for this purpose have been developed (e.g., Capitainer-B or HemaPEN^TM^). Thanks to the ease of sample collection and storage, DBS thrive in research and pre-clinical studies, forensic toxicology, as well as clinical analysis. In fact, DBS have been employed for the newborn screening of inborn errors of metabolism for over half a century and a variety of other clinical applications including endocrinology, toxicology, and therapeutic drug monitoring have been implemented [30,31].

The present study describes a DBS sampling method based on an on-spot derivatization of GSH. This method enables the accurate determination of GSH and GSSG by ultra-performance LC (UPLC)-MS/MS, and overcomes GSH stability and storage issues. Results obtained from the analysis of >150 whole blood samples from newborn infants show its suitability for routine clinical laboratory determinations as well as for its use in clinical trials. Furthermore, the reference ranges of GSH and GSSG for healthy term infants are established.

## 2. Materials and Methods

### 2.1. Standards and Reagents

GSH, GSSG, GSH-(*glycine*-^13^C_2_,^15^N) trifluoroacetate salt, NEM and acetonitrile were purchased from Merck Life Science (Madrid, Spain). GSSG-D_10_ was provided by Toronto Research Chemicals (Toronto, ON, Canada). Formic acid (FA) (98%, *v*/*v*) and perchloric acid (PCA) (70%, *v*/*v*) were purchased from Panreac Química (Barcelona, Spain). Phosphate buffer solution (PBS) tablets were from VWR (Barcelona, Spain). The preparation of stock, working, and standard solutions is described in the Appendix A.

### 2.2. UPLC-MS/MS Chromatographic System

A Waters Acquity UPLC-Xevo TQD system (Milford, MA, USA) was employed for UPLC-MS/MS analysis. The mass detector was operated in multiple reaction monitoring mode. Analytical parameters, summarized in Appendix A), were adapted from [20]. Ionization and fragmentation parameters for GSSG-D_10_ detection were selected by the direct infusion of the analytical standard into the ESI-MS/MS system.

### 2.3. Sample Collection and Processing

Filter paper cards employed for newborn screening were cut with a paper puncher and 6 mm diameter papers were obtained. Individual papers were introduced in separated Eppendorf tubes to which 20 μL of a 200 mmol/L NEM solution in PBS were added. Then, papers were dried under vacuum (30 °C, 45 min) and the impregnation-drying process was repeated once, giving a final amount of 8 µmol of NEM. NEM-treated papers were stored at 4 °C until their use.

Ten microliters of blood samples were directly added to the papers pre-soaked with NEM and allowed to derivatize (GSH-NEM formation) at room temperature (RT) for 5 min. Then, they were kept at least 3 h and up to 24 h at 4 °C until further processing. Then, 100 μL of cold PCA solution (4%, *v*/*v*) were added to the precipitate proteins. To extract analytes of interest, Eppendorf tubes were placed in a thermal mixer for 5 min at 4 °C at 1400 rpm followed by 5 min of sonication. Finally, samples were centrifuged at 10,000× *g* for 15 min at 4 °C, and 40 μL of supernatants were diluted with 60 μL of H_2_O (0.1% FA, *v*/*v*) and 5 μL of internal standard (IS) solution containing isotopically labelled GSH-NEM and GSSG at 100 μmol/L each. Thereafter, samples were stored at −80 °C until analysis. Paper card treatment with NEM and the sample collection and processing scheme are represented in Figure 1.

### 2.4. Method Performance

In accordance with the guidelines for setting up DBS-based methods for Therapeutic Drug Monitoring [32], the DBS method was compared with a previously developed method [20] for GSH-NEM and GSSG determinations in liquid blood samples (see Appendix A). Thus, 28 blood samples were analyzed by both methods and side-by-side comparison via Passing–Bablok regression [33] and Bland–Altman [34] analysis was performed. In addition, GSH-NEM and GSSG stability on DBS papers was studied in blood samples testing three different conditions (RT, 4 °C and −20 °C) from 1 to 28 days.

### 2.5. Blood Samples and Clinical Approach

Cord blood samples (N = 73) and residual blood volumes from routine newborn screening performed at discharge (N = 88) from healthy term infants were collected in EDTA-containing tubes. In terms of sampling and storage, two different clinical scenarios were addressed: short- and long-term storage (STS and LTS, respectively). The first scenario (STS) represented the routine clinical practice of daily determinations in blood, where DBS samples are collected and processed after storage at 4 °C for 24 h. For the second scenario (LTS), collected DBS were stored at −20 °C and processed within one month of collection, simulating the scenario of clinical trials, where samples are usually collected individually but processed in batches. The agreement between GSH-NEM and GSSG concentrations in both simulated clinical scenarios was evaluated by linear regression and by the Bland–Altman plot [34] using a set of 99 DBS tests carried out in duplicate for STS and LTS.

## 3. Results

### 3.1. Derivatization Efficiency Using NEM-Treated Papers

Individual papers were treated with increasing amounts of NEM (i.e., 2, 4, 5, 6, 8, and 10 μmol). After drying under vacuum, papers pre-soaked with NEM were stored at 4 °C until the next day. Appendix A) shows GSH-NEM and GSSG values of the same whole blood sample deposited on DBS paper cards pre-soaked with increasing amounts of NEM. As can be seen, GSSG values reached a plateau when ≥8 μmol of NEM were employed. Then, the efficiency of GSH derivatization, a critical step to avoid GSH oxidation, was evaluated by adding increasing amounts of GSH (0.25–2 mmol/L) onto DBS paper cards pre-soaked with NEM. As shown in Figure 2, the derivatization successfully occurred under the assayed conditions. On the other hand, when a whole blood sample was processed using papers without NEM, a decrease in GSH-NEM (0.90 ± 0.03 vs. 0.20 ± 0.01 mmol/L) and a huge concomitant increase in GSSG were observed (3.1 ± 0.3 vs. 276 ± 18 µmol/L), thus confirming the need for an immediate derivatization of GSH with NEM upon sample collection.

### 3.2. Method’s Performance

As shown in Appendix A), the LC-MS analysis provided an adequate linearity for GSSG within the corresponding concentration range (9–9400 nmol/L; R^2^ = 0.997), whereas for GSH-NEM, quadratic curve fitting was more suitable (185–190,000 nmol/L; R^2^ = 0.9996). LODs and LOQs, estimated as the concentration providing a signal-to-noise ratio of 3 and 10, respectively, were also in agreement with previously reported ranges [20,35].

The impact of the time elapsed between the collection of blood samples (t = 0) and sample processing (placing 10 μL of blood on top of papers pre-soaked with NEM) was evaluated at 10, 20, 30, 60, 120, and 180 min (N = 3). Relative variation in GSH-NEM and GSSG values with respect to t = 0 within this time range (10–180 min) ranged between −10 and 1% for GSH-NEM and between −8 and 10% for GSSG (see Appendix A).

GSH-NEM and GSSG determination with modified DBS tests was compared with a previously developed method for liquid blood samples [20]. First, and as a condition for Passing–Bablok regression analysis (see Figure 3), a linear relationship between methods was assessed using the *cusum* test.

The regression coefficients with a 95% confidence interval (CI) for the Passing–Bablok analysis were 0.97 [0.82, 1.12] and 1.24 [0.90, 1.66] for the GSH-NEM and GSSG slope, respectively; and −0.14 [−0.28, 0.002] and −0.02 [−0.18, 0.11] for GSH-NEM and GSSG intercept, respectively.

The Bland–Altman difference plot with log-transformed data was employed to assess the agreement between both methods and estimate bias (see Appendix A). This plot assesses the agreement between two quantitative methods of measurement through the evaluation of the bias between the mean differences. It estimates an agreement interval, within which, e.g., 95% of the differences of the second method, compared to the first one, fall [36].

To assess the stability of GSH-NEM and GSSG on papers pre-soaked with NEM, three different conditions of storage (RT, 4 °C, and −20 °C) from 1 to 28 days were assayed. Figure 4 shows the GSH-NEM (left) and GSSG (right) relative variations of the same blood sample following the proposed method. As can be seen, the temperature and time have an impact either on stability or on subsequent extraction. A decreasing trend for GSH-NEM with time was observed, whereas for GSSG, no evident trend could be seen. The strongest effect on GSH-NEM determinations was observed, however, with temperature, especially if storage occurred at RT.

The effect of the storage of NEM-treated papers on the efficiency of derivatization was also assayed. The obtained results show that no detrimental effect on their performance after one year of storage at 4 °C occurred when compared with NEM-treated papers kept for one single day (*p*-values > 0.05 for GSH-NEM (0.58 ± 0.06 vs. 0.60 ± 0.07 mmol/L) and GSSG values (0.23 ± 0.08 vs. 0.29 ± 0.09 µmol/L); Wilcoxon rank sum test; N = 3).

### 3.3. GSH Reference Ranges in Newborn Infants

The reliability of the developed method was tested by analyzing cord blood samples (N = 73) and residual blood volumes in EDTA-containing tubes from routine newborn screening at discharge (N = 88) from healthy term infants. The clinical information of the samples is shown in Appendix A.

Among the set of 161 analyzed DBS samples, only one was classified as a potential outlier and removed from further analysis since its GSSG value was 100 times higher than the median value (1.1 μmol/L). Neither GSH nor GSSG concentrations followed a normal distribution (Kolmogorov test, *p*-value < 0.001). Figure 5 shows the GSH-NEM, GSSG, and GSH-NEM/GSSG of cord blood and blood samples collected at discharge. GSH-NEM and GSSG median values (interquartile range, IQR) for cord blood samples were 0.92 (IQR = 0.28) mmol/L and 2.4 (IQR = 1.8) μmol/L, respectively, and were significantly lower than those found in samples collected at discharge (i.e., 1.31 (IQR = 0.42) mmol/L GSH-NEM, *p*-value < 0.001; 3.9 (IQR = 2.1) μmol/L GSSG, *p-*value < 0.001; Wilcoxon rank sum test). However, the GSH-NEM-to-GSSG ratios were not statistically different between both groups (*p*-value > 0.05; Wilcoxon rank sum test).

Correlations between clinical parameters included in Appendix A and GSH-NEM, GSSG, and GSH-NEM/GSSG were assessed (Pearson correlation). In cord blood samples, a moderate degree of correlation between GSH-NEM values and hematocrit (r = 0.46; *p*-value < 0.001) was observed—whereas the GSH-NEM-to-GSSG ratio was positively correlated with gestational age (r = 0.45; *p*-value < 0.001) and birth weight (r = 0.40, *p*-value < 0.001) (see Appendix A). The possible effect of the mode of birth on GSH and GSSG values in cord blood samples was also inspected. In samples corresponding to vaginal delivery (N = 17), GSH-NEM and GSSG values were higher than in C-section samples (N = 56) (1.1 (IQR = 0.3) > 0.9 (IQR = 0.2) mmol/L GSH-NEM; 3.0 (IQR = 1.8) > 2.1 (IQR = 1.8) μmol/L GSSG), with a significant difference for GSH-NEM (*p*-value < 0.05; Wilcoxon rank sum test). For samples collected at discharge, no noticeable correlation with clinical parameters was observed.

### 3.4. Assessment of Two Clinical Scenarios

GSH-NEM and GSSG results for two clinical scenarios were examined involving STS (4 °C for 24 h) and LTS (−20 °C for 1 to 30 days), simulating the typical conditions observed in daily determinations in routine blood tests and sample collection in clinical trials, respectively. Figure 6 depicts the results obtained for GSH-NEM and GSSG. No outlier (i.e., values with more than three scaled median absolute deviations away from the median) or clear discrepancies (i.e., noticeable differences between STS and LTS results) were observed for GSH-NEM, whereas for GSSG, two and five outliers for STS and LTS, respectively, were detected and significant differences between STS and LTS were observed for three data points. Hence, for GSSG, three data points were removed before further data analysis. Regression analysis with data fitted to a linear model is depicted in Figure 6a,b (top). Correlation coefficients (r) between the two storage conditions for GSH-NEM and GSSG results were 0.76 (*p*-value < 0.001) and 0.77 (*p*-value < 0.001), respectively. Linear curve equations with a confident interval (CI) (α = 0.01) were GSH-NEM_LTS = [0.59, 0.94]·GSH-NEM_STS + [0.10, 0.52], and GSSG_LTS = [0.67, 1.1]·GSSG_STS + [0.25, 1.70]. In addition, to assess the differences between storage conditions, the differences between the results obtained under both conditions against their mean were represented in a Bland–Altman plot [34] (Figure 6c,d). For GSH-NEM, the mean (bias) and standard deviation of differences were −0.05 and 0.20 mmol/L, respectively, and the limits of agreement were comprised between −0.45 to 0.36 mmol/L. For GSSG, the mean (bias) and standard deviation of differences were −0.5 and 1.2 μmol/L, respectively, and the limits of agreement were comprised between −3.0 and 1.9 μmol/L. Appendix A) shows the median and IQR values of GSH-NEM, GSSG and the GSH-NEM-to-GSSG ratio of cord blood samples and samples at discharge stored and processed under both conditions.

## 4. Discussion

Modified DBS arises as a valuable alternative for redox status monitoring. The use of filter papers pre-soaked with 8 μmol of NEM for sampling for an immediate sample derivatization provided adequate analysis performance for both GSH-NEM and GSSG (see Figure 2 and Appendix A). The side-by-side comparison of the DBS-method vs. liquid blood analysis showed that both methods were statistically comparable for GSH-NEM and GSSG determinations (see Figure 3). However, the confidence interval for the slope in GSSG as well as the span of the limit of agreement in the Bland–Altman analysis (see Appendix A) were wide. This might be due to the low GSSG concentrations found in the samples, which are close to the limit of quantification.

Since GSH derivatization directly occurs upon sample collection, our procedure allows an accurate determination of GSH-NEM and GSSG with minimal manipulation by clinical staff. Moreover, this strategy could be further implemented in microfluidic devices such as Capitainer-B or HemaPEN^TM^, allowing the volumetric collection of blood circumventing the use of pipettes. Regarding the sample collection, NEM-treated papers were stable for at least one year and stored conveniently at 4 °C, thus facilitating the practical implementation of the method. GSH-NEM and GSSG in blood samples were stable for at least 3 h upon collection in EDTA-containing tubes before being deposited on NEM-treated papers (see Appendix A). It is worth noting that the DBS method could also be employed for blood samples collected without the addition of anticoagulants. In this case, an immediate deposition of the sample on the filter paper would be required. With respect to DBS test stability, after storage at RT, DBS-papers appeared darker than at 4 °C and −20 °C. In addition, considering that GSH derivatization with NEM in aqueous media is not hampered at RT, the decreasing trend observed for this compound, especially at longer times (see Figure 4), was attributed to an inefficient extraction in the subsequent steps. This was confirmed by the absence of a concomitant increase in GSSG. The effect that the temperature and derivatization time had on GSSG did not follow a monotonic upward or downward trend. Concentrations of GSSG and GSH in blood are in the micromolar and millimolar range, respectively. Thus, the oxidation of even small percentages of GSH during sample collection, storage or pre-processing leads to a significant increase in GSSG, introducing bias in the results. The results depicted in Figure 4 indicate that storage at 4 °C is acceptable for 24–48 h storage times. However, for longer times, storage at −20 °C is required.

The analysis of 161 blood samples from newborn infants demonstrated the suitability of the proposed methodology. Only one single sample was classified as an outlier. This proves the potential of our method for its application in clinical trials and routine determinations. Reference ranges for GSH-NEM and GSSG concentrations, as well as the GSH-NEM/GSSG ratio could be established for term infants at birth and at hospital discharge (see Figure 5). The results obtained for GSH are in the range of those previously reported for cord blood samples [7] and for residual blood volumes from routine newborn screening of healthy term infants [24]. Regarding GSSG, our results are lower than those reported for term infants [37] due to reduced GSH oxidation during the proposed sample collection protocol compared to previous methods without direct in situ GSH derivatization. The accuracy and precision provided by the method revealed associations between these compounds with several clinical parameters.

We assessed GSH-NEM and GSSG determinations for two typical clinical scenarios that reproduce the conditions of sampling and storage relevant for routine clinical analysis (STS) and clinical trials (LTS). GSH-NEM and GSSG concentrations after STS and LTS were significantly correlated (see Figure 6). On average, LTS provided slightly higher concentrations for both compounds as compared to STS. However, only clinical goals ultimately define whether the agreement interval is adequate for the purpose of a study. Considering the possible confounding variables associated with clinical samples jointly with the technical difficulty of accurate GSSG determination, we recommend the use of either STS or LTS conditions within the same study.

## 5. Conclusions

The developed DBS approach for the direct derivatization of GSH facilitates sample collection by clinical staff while preserving sample integrity. While the quantification of GSH-NEM and GSSG through LC-MS requires specialized personnel and facilities, its implementation in clinical studies is feasible through collaboration between clinical staff and analytical laboratories. Additionally, the small blood volumes required enable serial determinations in excess blood volumes withdrawn for other determinations. The method provides accurate and precise results and proved to be robust even when employed in a clinical environment outside of the research laboratory. We hope that this approach will pave the way for highly reproducible GSH and GSSG determinations in blood samples.

## Figures and Tables

**Figure 1 antioxidants-11-01165-f001:**
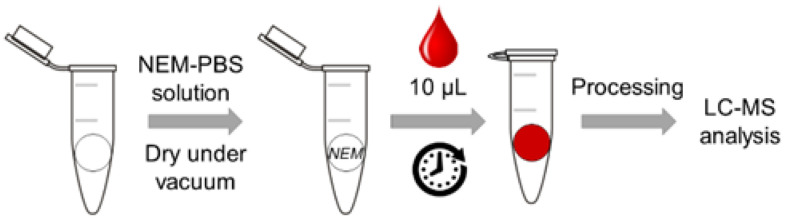
Paper card treatment with NEM, sample collection and processing scheme.

**Figure 2 antioxidants-11-01165-f002:**
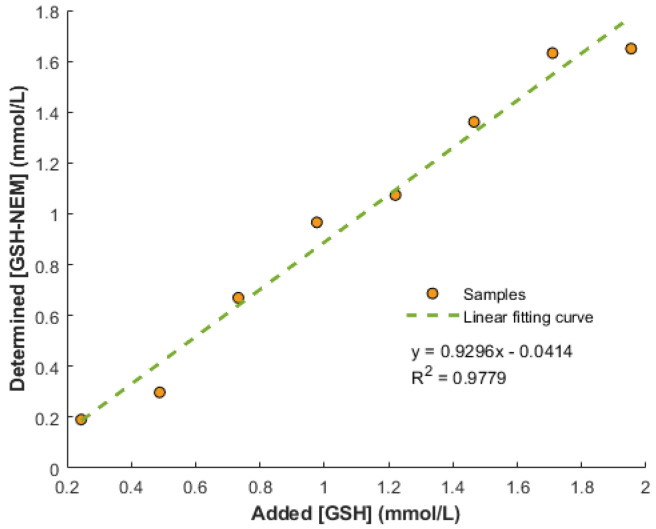
Experimentally determined GSH-NEM concentrations on DBS paper cards pre-soaked with NEM versus theoretical GSH concentrations of standard solutions.

**Figure 3 antioxidants-11-01165-f003:**
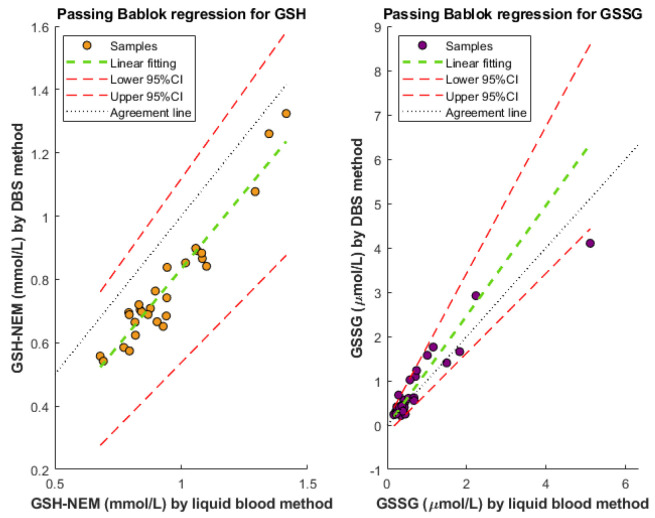
Passing–Bablok regression for GSH-NEM (**left**) and GSSG (**right**) for the comparison between the proposed DBS method and a previously developed method for liquid blood samples.

**Figure 4 antioxidants-11-01165-f004:**
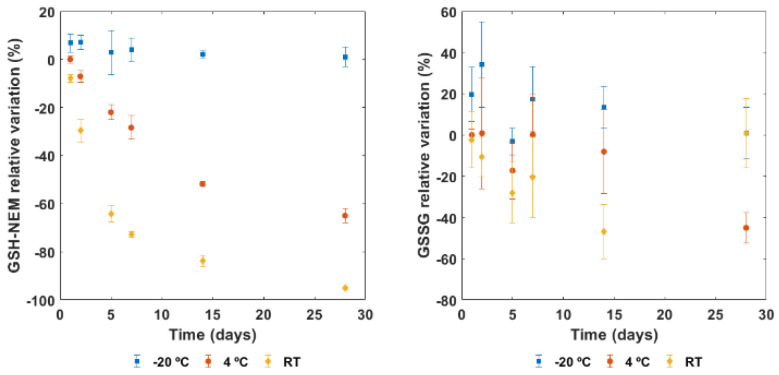
GSH-NEM and GSSG relative variation testing three different conditions (room temperature (RT), 4 °C, and −20 °C) from 1 to 28 days.

**Figure 5 antioxidants-11-01165-f005:**
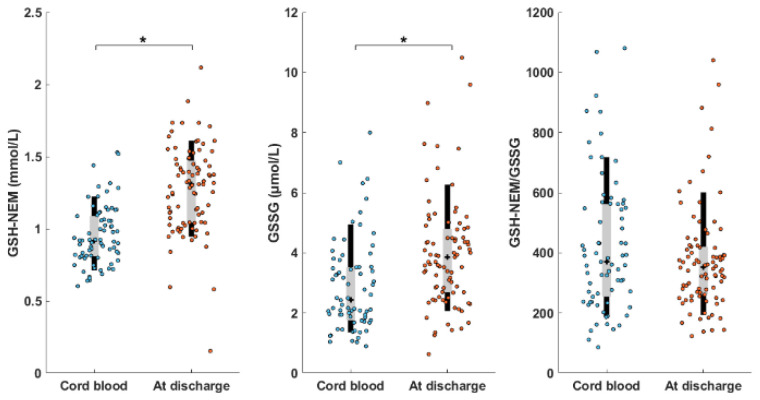
GSH-NEM and GSSG concentrations, and the GSH-NEM/GSSG of cord blood and blood samples collected at discharge. Black bars represent 10th–90th percentile; grey bars represent interquartile range (I–III), + represents median values. Note: * indicates *p*-value < 0.001; Wilcoxon rank sum test.

**Figure 6 antioxidants-11-01165-f006:**
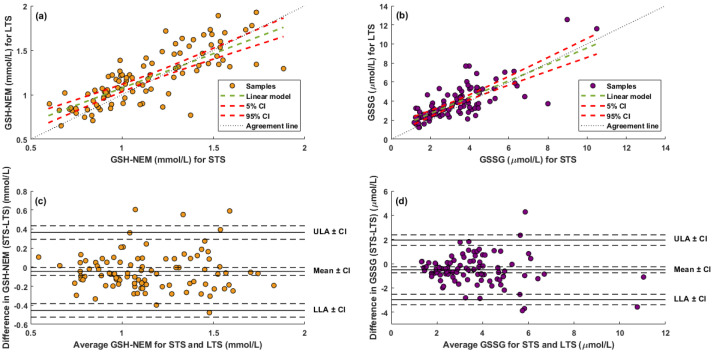
Regression line between GSH-NEM (**a**) and GSSG (**b**) concentrations measured under short- and long-term storage conditions (STS and LTS, respectively). Bland–Altman plot for GSH-NEM (**c**) and GSSG (**d**) comparing STS and LTS conditions. Note: CI, 95% confidence interval; LLA, lower limit of agreement; ULA, upper limit of agreement.

## Data Availability

Raw data are available from the authors upon request. The data are not publicly available due to privacy/ethical restrictions.

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
