# Peer review of "Direct Derivatization in Dried Blood Spots for Oxidized and Reduced Glutathione Quantification in Newborns"

_antioxidants, 2022, doi:10.3390/antiox11061165_

Round 1
Reviewer 1 Report
The present manuscript describes a novel method to determinate GSH and GSSG in dried blood spot 18 (DBS) samples based on the direct in-situ GSH derivatization with NEM. The manuscript is well written, the objective of this work is clear and results seems to be consistent. Although the discussion is missing, there are not a single reference which comparing results in all the discussion section. Authors must discuss the relevance of their results and the novelty comparing with others similar methods. Also I not agree with the conclusion of these author, this method can be easy to handle but need a LC-MS/MS to carry it out. This is a very expensive technique for clinical trials that need very specialized staff. This method is easy to fit in a research unit that could collaborate making these analysis. Thus, authors must change the conclusion of this manuscript and rewrite the discussion section.
Also, there are two references in the reference section (32 and 33) that are missing in the text.
Reviewer 2 Report
The authors developed a method to measure GSH and GSSG using dried blood spot (DBS) samples and confirmed that the results were comparable to those of conventional methods. In neonatal care, it is very important to perform necessary tests with a small amount of blood. In this regard, the development of a method to detect the GSH/GSSG ratio with a small amount of blood is of great significance in understanding the state of oxidative stress in neonates. In order to accept this paper, it would be helpful to add a little more explanation about the significance of measuring the GSH/GSSG ratio in the neonatal period in the introduction section.
Reviewer 3 Report
after careful reading of the subject manuscript I have no doubt about recommendation for publication in the present form. Authors satisfied all the propositions and above all presented a very well implemented development and validation of method for determination of GSH/GSSG in dried blood spots. The conducted study is of great clinical importance beacuse oxidant/antioxidant status has an ubiqutous role in disease diagnosis.
Round 2
Reviewer 1 Report
After the carefully revised version of the present manuscript my recommendation is accept in the present form